# Genome-wide systematic identification of methyltransferase recognition and modification patterns

Torbjørn Ølshøj Jensen[1], Christian Tellgren-Roth[2], Stephanie Redl[1,3], Jérôme Maury[1], Simo Abdessamad Baallal Jacobsen[1], Lasse Ebdrup Pedersen [1] & Alex Toftgaard Nielsen [1]

Genome-wide analysis of DNA methylation patterns using single molecule real-time DNA sequencing has boosted the number of publicly available methylomes. However, there is a lack of tools coupling methylation patterns and the corresponding methyltransferase genes. Here we demonstrate a high-throughput method for coupling methyltransferases with their respective motifs, using automated cloning and analysing the methyltransferases in vectors carrying a strain-specific cassette containing all potential target sites. To validate the method, we analyse the genomes of the thermophile *Moorella thermoacetica* and the mesophile *Acetobacterium woodii*, two acetogenic bacteria having substantially modified genomes with 12 methylation motifs and a total of 23 methyltransferase genes. Using our method, we characterize the 23 methyltransferases, assign motifs to the respective enzymes and verify activity for 11 of the 12 motifs.

---

[1] The Novo Nordisk Foundation Center for Biosustainability (CfB), Technical University of Denmark (DTU), DK-2800 Lyngby, Denmark. [2] Uppsala Genome Center, National Genomics Infrastructure, SciLifeLab, SE-751 08 Uppsala, Sweden. [3] Massachusetts Institute of Technology, Cambridge, MA 02139, USA. Correspondence and requests for materials should be addressed to A.T.N. (email: atn@biosustain.dtu.dk)

Modification of DNA with a methyl group, catalysed by DNA methyltransferases, is considered to be the most abundant form of post-replicative nucleotide modification found in genomes of both prokaryotes and eukaryotes[1,2]. In mammalian cells, DNA methylation is known to be involved in a wide range of functions in the regulation of the genome[3–5]. Methylation of DNA by methylatransferases in bacteria and archaea has mostly been described as part of the sequence-specific restriction modification system (R-M), which traditionally has been viewed as primitive immune systems of bacterial cells to withstand invasion of foreign DNA[6–11]. However, this conception has been progressively broadened to include additional roles/functions (recently reviewed by Vasu[12]). Along the R-M system, methylation expands the coding capacity of DNA to include distinction of original from newly synthesized DNA as well as it participates in regulating chromosome replication, nucleoid segregation, conjugation[13,14], and global gene expression regulation[1,15–17]. Following the nomenclature of R-M systems, which is based on functional arrangement and the requirement for cofactors and/or specific subunits[18,19], methyltransferases are classified in three main types (Type I, II, III). In bacteria, methyltransferases typically transfer a methyl group from a donor such as S-adenosyl-L-methionine (SAM) to a target nucleotide base, thereby forming N-6-methyladenine (m6A), N-4-methyl-cytosine (m4C), or C-5-methylcytosine (m5C)[20]. A considerable amount of information regarding both methyltransferases and restriction endonucleases exists[21], based on laborious experimental approaches such as radioactive labeling with [3H] S-adenosylmethionine[22], sensitivity to cleavage by restriction endonucleases[23], and methylation-specific PCR of bisulfite-modified DNA[24]. Recently, genome-wide analyses of methylomes have been published, relying on Single Molecule Real-Time DNA sequencing (SMRT) for direct detection of methylated DNA nucleotides[25], thereby making it possible to recognize all three types of bacterial methylation[2]. Other sequencing methods, including bisulfite sequencing[26,27], Nanopore technology, also have the potential to detect modifications of DNA[28,29]. SMRT sequencing has even been used to characterize methyltransferases at an individual level[2]. In 2016, Blow et al.[30] assigned motifs to multiple methyltransferases based upon significant similarities to methyltransferases with known specificity and methylome data (available at REBASE[31]). With the increasing number of methylomes publicly available (834 distinct methylation motifs were identified in 230 bacterial and archaeal species[30]) and relative easy generation of methylome data (currently still requiring human analysis of the data), the diversity of methylation between species and strains as well as mobility of RM systems or target recognition domain (TRD) has become apparent[32,33]. Approaches comparing methylomes and genomes to couple TRD of methyltransferases to specific patterns have been published, enabled by a specific degree of diversity and/or size of the compared groups[34,35]. Deciphering sequence specificity of methyltransferases not depended on information and specific diversity in a group of closely related strains still remains a challenging and laborious task. Algorithms predicting DNA-protein interactions can be applied to facilitate the coupling, but the accuracy of the predictions is typically only around 80%[36]. It is foreseen that experimental evidence could improve these algorithms significantly.

The present study links the DNA methylation pattern identified with SMRT sequencing in a high-throughput manner to sequence-specific enzymatic activities of the assessed methyltransferases. This is achieved by an automated design of strain-specific motif-cassettes containing all potential target sites, cloning of methyltransferases in vectors carrying the cassette, and easy sorting and handling of the data. Analysing methyltransferases from the thermophile *Moorella thermoacetica* and the mesophile *Acetobacterium woodii*, two acetogenic bacteria having substantially modified genomes, demonstrates the applicability and versatility of the method. Using this approach, we characterize 23 methyltransferases, assign motifs to the respective enzymes and verify activity for 11 of the 12 motifs.

## Results

**Genomic sequence and methylome data.** The two acetogenic strains, *M. thermoacetica* (thermophilic) and *A. woodii* (mesophilic), were selected for validating the developed method, since both are distant to the expression host *E. coli* and have different growth optima. Furthermore, both strains are gas-consuming bacteria with a scarce product spectrum, making them highly interesting hosts for production of biochemicals. However, genetic engineering of the two organisms is still challenging, most likely due to foreign DNA being digested by the native R-M systems.

Initially, the genome and methylome of *M. thermoacetica* was determined using SMRT sequencing technology, with an average coverage of 137-fold and mean mapped subread length of 11,250 bp using parameters described in the methods section. Previously, similar genome sequencing of *A. woodii* resulted in an average coverage of 163-fold[30]. The genomes of the strains are relatively small, 2.63 Mb for *M. thermoacetica* (accession number CP031054) and 4.04 Mb for *A. woodii* (accession CP002987)[37]. The two strains showed significant differences in their methylomes. The thermophilic acetogen *M. thermoacetica* had a high modification frequency of 22 modifications/kb (modifications on either strand), corresponding to every 45th nucleotide on average being modified, whereas *A. woodii* only had a frequency of 4.4 modifications/kb (Table 1). Modification frequencies of *M. thermoacetica* is comparable to the most heavily methylated genomes analysed so far by SMRT technology[30,38–40]. Average occurrence of three motifs per organisms (in bacteria or archaea) has been observed[30]. In *M. thermoacetica* seven motifs were identified, five m6A and two m4C modifications, only reads longer than 10 kb were considered and min QV of 85. The motifs included two asymmetric bipartite sequence motifs (characteristic for Type I methyltransferases), these appeared to complement each other, both modified at the first adenine (Table 1). One of the motifs included a "W" as first base, which was not included in the complement motif. The "W" was considered miscalled caused

**Table 1 Summary of methylome data of the two different strains**

| Motifs | Modified position | Type | % motifs detected | # of motifs in genome |
|---|---|---|---|---|
| *M. thermoacetica* 39073-HH | | | | |
| WATCNNNNNCTC | 2 | m6A | 100.0% | 461 |
| GAGNNNNNGAT | 2 | m6A | 99.4% | 950 |
| SATC | 2 | m6A | 99.9% | 37470 |
| GAWTC | 2 | m6A | 99.7% | 4600 |
| AACCA | 5 | m6A | 99.7% | 5547 |
| GGGCCC | 5 | m4C | 97.4% | 2352 |
| CTCCG | 4 | m4C | 76.4% | 5439 |
| *A. woodii* DSM 1030 | | | | |
| GCCRAG | 5 | m6A | 99.6% | 1989 |
| CCWGG | 1 | m5C | – | 9414 |
| TAAGNNNNNTCC | 3 | m6A | 99.8% | 304 |
| GATGNNNNNNTGC | 2 | m6A | 99.8% | 375 |
| CAAAAAR | 6 | m6A | 98.9% | 5576 |

The data for *A. woodii* was from Blow et al.[30]

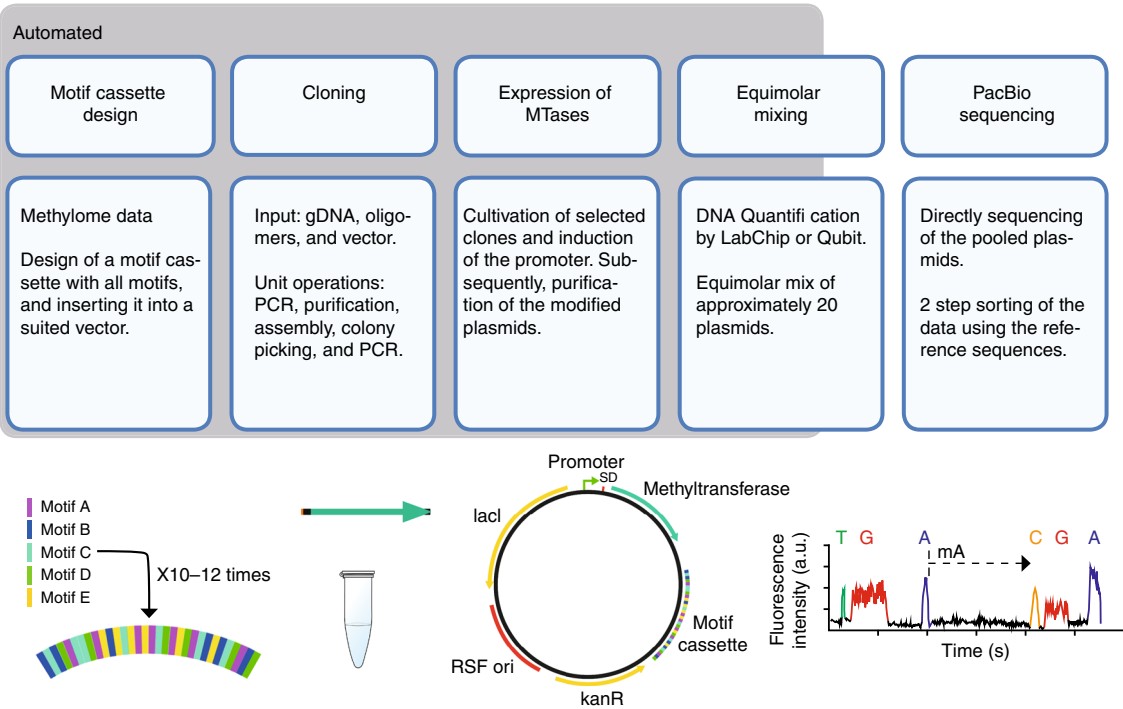

**Fig. 1** MetMap workflow. The box designated "automated" indicates the steps that were automated. The variance in the occurrence of motifs in the motif-cassette depends on the nature of the motifs; A motif including unspecific regions of 2–10 n's (typical for Type I motifs) is present 12 times in the cassette, whereas motifs not including such regions is repeated 10 times

by the overlapping motif SATC modified similarly at the adenine base. SATC was the most frequently occurring motif in *M. thermoacetica*, occurring 37,470 times in the genome. Furthermore, two non-palindromic motifs were detected, which are typical for Type III and some Type II methyltransferases.

In *A. woodii* five motifs were observed (PacBio data from previous study[30]); four m6A and one m5C modification, none of these were identified in *M. thermoacetica*. The motifs of *A. woodii* included two asymmetric bipartite sequence motifs. The palindromic sequence CCWGG, m5C modified at the first position, was the most frequently occurring motif appearing 9414 times in the genome. The remaining two motifs appeared in total 7565 times, were non-palindromic expressing high specificity with only one degenerate nucleotide in each motif. For *A. woodii*, methyltransferase genes encoded in the genome had be been linked to four of the five motifs based on similarities to other already characterized methyltransferases and deduction[30]. Similar bioinformatic analysis of motifs was performed on the genes of *M. thermoacetica*; motifs were predicted for three genes but deviating from the methylome identified by SMRT sequencing. In the two strains, nine modification motifs were assigned to genes using the pipeline at REBASE[31]. However, of the 23 methyltransferase genes more than 60% had unknown target sites. More importantly, a considerable number of modifications identified were not assigned to specific genes. This emphasizes the need for generating more experimental data to support the methyltransferase specificity predictions.

**Motif cassette design and cloning of methyltransferase genes**. Coupling the methyltransferases to the specific motifs determined by Pacbio-sequencing of the genomes, as described above, would typically require a tedious workflow including cloning, expression, and individual activity determination. The method developed in this study was designed to enable a high-throughput workflow and carry out the sequencing in a single library without

barcoding (Fig. 1). Central to the high-throughput workflow was the design of the methylome-specific motif cassette. The motif cassettes included all motifs randomly organized, assuring that copies of the same motifs were not placed directly adjacent to each other. Motifs including unspecific regions of 2–10 n's (typical for Type I motifs) were present 12 times in the cassette, whereas motifs not including such regions were repeated 10 times (Fig. 1). Whenever motifs included degenerate nucleotides, all possible combinations were integrated as independent motifs. This construct enables detection of all motifs solely based on the motif cassette, not relying on the presence of motifs on the plasmid backbone.

A script, MetMap, was developed to facilitate the construction of the motif cassette. The script automates the design based on the motifs identified by SMRT sequencing of the genomes with the prerequisites listed above and can be accessed at https://github.com/biosustain/metmap, along with guidelines and documentation. The motif-cassettes for the two evaluated strains, created by MetMap, were synthesised and placed in the expression vector having the methyltransferase gene and corresponding terminator inserted upstream and a NotI site downstream. The NotI site was used to linearize the plasmids, thereby enabling the construction of full-length plasmid sequencing libraries. The 23 methyltransferases from *M. thermoacetica* and *A. woodii* were cloned into vectors carrying the strain-specific motif cassettes (Supplementary Figs. 4, 5) using an automated cloning platform. Plasmids holding and expressing the methyltransferases were subsequently linearized and pooled in equimolar amounts prior to PacBio sequencing.

**SMRT-seq of pooled plasmids expressing single methyltransferases**. Eleven and fifteen different plasmids with individual methyltransferases were pooled and sequenced using one SMRT-cell (a pool of up to 19 plasmids were sequenced in one SMRT-cell with sufficient coverage). Sequences of the individual

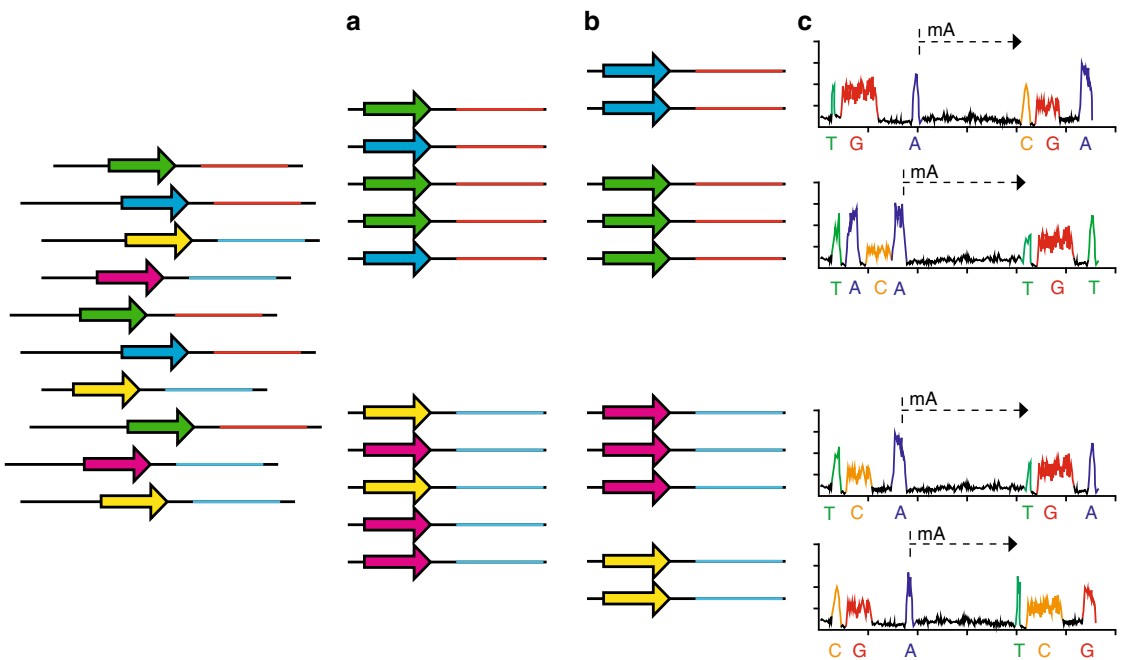

**Fig. 2** Handling of the sequences. **a** mapping based selection of the sequences based upon the motif cassette; **b** 2nd round of mapping based selection based upon both the motif cassette and the gene coding for the methyltransferases; **c** analysing the grouped sequences though the 'Modification and Motif_Analysis' pipeline

methyltransferases and negative controls were isolated using a two-step sorting process (outlined in Fig. 2): initial sorting based only on the strain-specific motif cassette, followed by a sorting where the reads were clustered according to the methyltransferase sequences and the motif cassette. Thereafter, the modification patterns of each single methyltransferase were determined.

Of the 23 methyltransferases and two negative controls, it was possible to isolate all coding sequences and identify 21 modification motifs with a detection rate close to 100% for all (Supplementary Data 1). Additional motifs were detected (Supplementary Data 1 lists all information from the motif_summary.csv files) under the given parameters, however upon inspection some appeared as miscalled motifs. The coverage ranged from 116- to 11,339-fold (using std. parameters), and we found no correlation between coverage and the DNA concentration or size (see Supplementary Fig. 1 and Supplementary Fig. 2). The large coverage span did not make it possible to analyze all samples using the standard parameters (min read score = 0.75, min modification QV = 30), as the scoring algorithm in the analysis pipeline is coverage sensitive. Additionally, detection of the methylation signal requires different read coverage depending on the modified base. Detection of m5C requires at least five- to 10-times more coverage than for m6A and m4C, due to lower kinetic variation signals[25,34,38,39,41–43]. Therefore for each sample the minimum read score was adjusted to reduce the coverage and increase the modification QV cut-off to avoid false positive identifications of highly degenerate motifs.

The identified modification motifs with the specific methyltransferase genes are summarized in Tables 2 and 3. Eleven of the modifications identified, have motifs corresponding to the motifs identified by the initial PacBio sequencing of the genomes (and included in the motif cassette design). As 12 motifs were identified initially, the efficiency was 92%. The only modification not properly assigned to a specific methyltransferase was the m4C methylation on the fourth position of the sequence CTCCG from *M. thermoacetica* (one methyltransferases indicated activity on the motifs). No methyltransferase activity was detected from the

negative controls. The expressed control genes were assessed for their functionality in the presence of 5-bromo-4-chloro-3-indolyl-β-D-galactopyranoside (x-gal)[44,45].

**Detection of motifs from different methyltransferase types**. To demonstrate the versatility of the method, methyltransferases annotated as both Type I, II, and III were included. One thermophilic Type I methyltransferase, encoded by the two genes MothHH_01799 (methyltransferase subunit) and MothHH_01798 (DNA specificity subunit), was included in the study. When heterologous expressed, two extended and complement motifs were detected, which could be linked to the motif ATCNNNNNCTC identified on the genome (if more Type I motifs were present in the genome this may be more challenging). Four mesophilic Type I methyltransferases were also included and expressed together with their corresponding specificity subunits (with the exception of Awo_c17840, which did not have an obvious specificity subunit located close to the gene). Two sets (Awo_c08800_Awo_c08810 and Awo_c17130_Awo_c17140) demonstrated activity on both strands, whereas Awo_c17840 (methyltransferase subunit) and Awo_c04420_Awo_c04450 did not show any activity. Looking into the lack of activity of Awo_c04420_Awo_c04450, the two open reading frames (ORFs) had sequence-similarities to specificity subunits (HsdS), suggesting that a mutation may likely have split the subunit into two potential nonfunctional ORFs. A mutation at position 522,352 (at the end of Awo_c04440) was identified by Blow et al.[30] and included as a comment on REBASE to explain why no modifications were expected. Motif identification of Awo_c17130_Awo_c17140 (at QV255) resulted in two motifs that appeared as complement, despite diminutive differences. Compared to the motif found in the methylome of *A. woodii*[30] an ambiguous nucleotide was present on the fourth position of the 5′-submotif, thus it is reasonable to link activity of the methyltransferases to motifs observed in the genomes. Awo_c17840 did not show any activity, which could be explained by the lack of a

**Table 2 Summary of the methyltransferases from *M. thermoacetica***

| Gene name ATCC 39073-HH | Gene name ATCC 39073 | Gene product | Predicted specificity | Motif detected | Mean mod. QV | Coverage |
|---|---|---|---|---|---|---|
| *Type I* | | | | | | |
| MothHH_01798 | Moth_1671 | DNA met. Specificity | N.A. | W<u>G</u>A$^{(m6A)}$ | 55.40 | 1643.71 |
| MothHH_01799 | Moth_1672 | N-6 DNA methylase | | *<u>GNNNN</u>AGATNNNNC | 53.80 | |
| | | | | <u>GNNB</u>NA$^{(m6A)}$ | | |
| | | | | *TCTNNNWNTC | | |
| *Type II* | | | | | | |
| MothHH_00029 | Moth_0026 | DNA adenine methylase | GASTC | GA$^{(m6A)}$WTC | 49.58 | 50.29 |
| MothHH_00692 | Moth_0639 | DNA methylase N-4/N-6 | N.A. | BGN<u>STSC</u>$^{(U.K.)}$ | 35.40 | 224.54 |
| | | | | <u>NNGN</u>NNRNR | | |
| MothHH_00927 | Moth_0871 | conserved hypothetical protein | N.A. | <u>C</u>A$^{(m6A)}$TC | 88.75 | 95.23 |
| MothHH_01869 | Moth_1737 | DNA adenine methylase | GGATC | GA$^{(m6A)}$TC | 53.14 | 60.88 |
| MothHH_02421 | Moth_2238 | hypothetical protein | N.A. | N.D. | N.D. | 6388 |
| MothHH_02424 | Moth_2241 | hypothetical protein | N.A. | N.D | N.D. | 1531 |
| MothHH_02467 | Moth_2281 | Adenine-specific DNA methylase | AGGCCT | GGGCC$^{(m4C)}$C | 151.5 | 343.27 |
| *Type III* | | | | | | |
| MothHH_01297 | Moth_1213 | Site specific subunit | N.A. | AACCA$^{(m6A)}$ | 69.93 | 83.43 |
| Neg-control | | bgaB (beta-galatosidase) | None | N.D. | N.D. | 116 |

Methyltransferases from *M. thermoacetica* cloned individually and their corresponding motifs
*N. A.* not available, *N. D.* not detected, *U. K.* unknown
*indicates that the methylation type was identified by the modification position

**Table 3 Summary of the methyltransferases from *A. woodii***

| Gene name (DSMZ 1030) | Gene product | Predicted specificity | Motif detected | Mean mod. QV | Coverage |
|---|---|---|---|---|---|
| *Type I* | | | | | |
| Awo_c04420 | R-M subunit HsdM1 | N.A. | N.D. | N.D. | 2075 |
| Awo_c04450 | R-M subunit HsdS1 | | | | |
| Awo_c08800 | R-M subunit HsdM2 | TAAGNNNNNTCC | TAA$^{(m6A)}$GNNNNNTCC | 1914 | 3446 |
| Awo_c08810 | R-M subunit HsdS2 | | GGA$^{(m6A)}$NNNNNCNTA | 1567 | |
| Awo_c17130 | R-M subunit HsdS3 | GATGNNNNNNTGC | GA$^{(m6A)}$TNNNNNNNTGC | 1136 | 1768 |
| Awo_c17140 | R-M subunit HsdM3 | | GCA$^{(m6A)}$NNNNNNCAT | 932 | |
| Awo_c17840 | R-M subunit HsdM4 | N.A. | N.D. | N.D. | 8535 |
| *Type II* | | | | | |
| Awo_c06460 | DNA (cytosine-5-)-methyltransferase Dcm | CCGG | N.D. | N.D. | 8569 |
| Awo_c14460 | DNA modification methyltransferase | N.A. | CAAAAA$^{(m6A)}$R | 2185 | 6372 |
| Awo_c18570 | Putative DNA methylase | N.A. | N.D. | N.D. | 9517 |
| Awo_c18590 | Putative DNA methylase | CCWGG | TNNN<u>CCTG</u>$^{(U.K.)}$*GNGNNV | 48 | 8514 |
| | | | MNN<u>BSNT</u>NNCCNG$^{(U.K.)}$*G | 32 | |
| Awo_c19740 | Potential Type II | GCCRAG | GCCRAG | 1986 | 6223 |
| Awo_c30860 | Putative DNA methylase | N.A. | TA$^{(m6A)}$GNHNNNNV | 608 | 5539 |
| | | | BCANA$^{(m6A)}$NNGNNNNNM | 172 | |
| | | | VNTNNNNNNTTNTA$^{(m6A)}$ | 147 | |
| Awo_c31130 | DNA (Cytosine-5-)-methyltransferase YdiP | N.A. | N.D. | N.D. | 11339 |
| Awo_c34920 | N6 adenine-specific DNA methylase | N.A. | WBNNMDVA$^{(m6A)}$GCNH | 70 | 937 |
| Awo_c35180 | phage N6 methylase | TGGCCA | WNNNNNMGNC$^{(m4C)}$ | 237 | 708 |
| | | | NGGNM | | |
| Awo_c35200 | Lambda C methyl | N.A. | N.D. | N.D. | 909 |
| Neg-control | bgaB (beta-galatosidase) | None | N.D. | N.D. | 8051 |

Methyltransferases from *A. woodii* cloned individually and their corresponding motifs
*indicates the modification position is misleading, as the modification on a guanine is not likely, in addition m5C has typical characteristic kinetic signals two and six bases downstream of the methylated position
*N. A.* not available, *N. D.* not detected, *U. K* unknown

specificity subunit. REBASE predicted motifs for both Type I methyltransferases of *A. woodii* (Awo_c08800_Awo_c08810 and Awo_c17130_Awo_c17140). The prediction was based on similarities of Awo_c08800 to S.Kaq16071III and deduction (excluding potential nonfunctional methyltransferases). Our results experimentally validate the predictions, and demonstrate the strength of this method to detect modification by Type I methyltransferases directly.

Of the seven Type II methyltransferases from *M. thermoacetica*, it was possible to assign motifs to four of them. Interestingly, the motif SATC identified on the genome turned out to be a double motif of GATC and CATC, modified by two different methyltransferases MothHH_01869 and MothHH_00927, respectively. The motif CTCCG was not properly identified by any of the cloned methyltransferases or the tested combination of TRD and methyltransferases, given the low detection rate on the genome sequencing (Table 1) this was expected. However, the motifs of MothHH_00692 indicated that it could have activity towards the CTCCG motif (extended motif including the motifs STSCN). Six of the ten cloned *A. woodii* methyltransferases displayed activity, three of which corresponded to the modification on the genome, and the remaining three were assigned to promiscuous motifs. Activity of Awo_c18590 resulted in recognition of four motifs, all with relative low modification QV (32–52), associating the motifs to the methylome, the activity could be assigned to motif CCWGG. Despite a coverage of 8514 (>250) modification QV was low and the motif extended, this indicates that detection of m5C by PacBio is incomplete. The motif for Awo_c19740 was identified as GCCRAG correlating with the predicted motif at REBASE (57% identity at protein level to *ateTI* from *Anaerococcus tetradius*). Furthermore, we identified the methyltransferases Awo_c14460 to be responsible for the CAAAAAR modification. Awo_c14460 did not have any predicted specificity. Awo_c30860, Awo_c34920, Awo_c35180, and M00910 were found to have unspecific activity, which could be because the target motifs might not have been included in the cassette, and were thus not present in sufficient number to determine the specific motifs. It could also be because the TRD or specificity subunit was not expressed simultaneously. It was observed that two of the genes Awo_c34920 and Awo_c35180 are encoded on prophage regions, and the genes might therefore not be active in the genome as prophage-encoded genes generally are transcriptionally inactive until the prophage is excised[46]. *M. thermoacetica* had one annotated Type III methyltransferase, its expression lead to the detection of a non-palindromic motif AACCA. No Type III methyltransferases were annotated in the genome of *A. woodii*. Despite different modes of action, both Type II and Type III methyltransferases can recognize non-palindromic motifs. This demonstrates that the developed method is also suitable for Type III methyltransferases.

**Restriction endonuclease digestion of modified plasmids**. In order to verify the modification motifs identified with the presented workflow, selected modifications were analysed by restriction digestion. The motifs that could be verified by digestion with commercial available restriction enzymes included GGGCCC, GATC, GAWTC, and CCWGG modified by the methyltransferases MothHH_02467, MothHH_01869, MothHH_00029, and Awo_c18590, respectively. Using the restriction enzymes ApaI, DpnI, PfeI, and EcoRII, all modifications were verified by hindrance or facilitation (DpnI) of digestion (Supplementary Fig. 3 and Supplementary Table 1).

**Comparison of predicted and detected motifs**. The sequence specificity of the methyltransferases predicted at REBASE is based on nucleotide sequence and similarities to characterized methyltransferases. Generally, the accuracy (percentage of correct prediction) of methods predicting protein-DNA binding sites are increasing but are usually about 70–90%[36]. By generating more experimental evidence, the specificity and accuracy of these models is expected to significantly increase. When comparing the motifs predicted by REBASE and those found experimentally in this study, some inconsistences were observed (see Tables 2 and 3). In total, nine methyltransferases had assigned predicted

specificity (REBASE), however only in four cases (all from *A. woodii*) were there a correlation with the specificity detected in this study. REBASE was able to predict three motifs for the methyltransferases of *M. thermoacetica*; GASTC, GGATC, and AGGCCT for MothHH_00029, MothHH_01869, and MothHH_02467, respectively. The present study determines the motifs to be GAWTC, GATC, and GGGCCC. Six methyltransferases from *A. woodii* had motifs assigned[30] (bioinformatic). Four of these correspond to four of the five motifs found in the methylome. In this study, the activity of the four methyltransferases was demonstrated and the methyltransferase responsible for the fifth motif, CAAAAAR was also identified.

## Discussion

In this study we presented MetMap, a novel method/pipeline to experimentally demonstrate the DNA modification specificity of methyltransferases in a high-throughput manner (summarized in Fig. 1). A total of 23 methyltransferases and a negative control (beta-galactosidase) were analyzed. Methylase activity was observed from 14 of the methyltransferases, and of 11 of these specificity could be related to their native activity on the genomes. The high throughput and scalability was facilitated by conducting the cloning steps using an automated cloning platform, but relying on gene amplification from genomic DNA. The advancements in gene synthesis methods have enabled assembly of large DNA fragments, with high fidelity at low costs. Exploiting newer lower-cost synthesis methods is expected to bring the cost further down[47]. In the future, such technology could effortlessly be implemented in the developed methods, and increase the throughput of our method even more.

The evaluated methyltransferases were derived from two different gram-positive acetogenic strains that are distant from the expression host *E. coli*, both with respect to growth temperature and codon usage. These strains were chosen specifically to demonstrate the robustness of the developed method. The methyltransferases were all expressed and evaluated without prior optimization, neither genes nor conditions. Heterologous expression of themophilic enzymes in mesophilic systems has been published, however, this often require either additional cofactors, appropriate posttranslational processing, or specific components during multisubunit assembly to enable production in an active form[48–52]. In this study, expression of the thermophilic methyltransferases at lower temperatures did not affect the activity of the proteins to a degree where it hampered the detection of the respective motifs. In fact, the frequency at which modification motifs could be linked to genes was even higher for methyltransferases derived from the thermophilic strain. Others have reported successful heterologous expression of thermostable methyltransferases in mesophlic systems[52,53], however it must be stressed that this observed compatibility of thermostable methyltransferases may not be general. Activity detection of the thermophilic Type I methyltransferases (multiple component enzyme) was arduous, as modQV was low and the motif slightly extended. This was in contrast to the high modQV and accurate motifs when assessing the Type I methyltransferases from the mesophilic host. These differences indicate that the thermophilic holoenzyme may have had reduced activity of reasons elaborated above, however, the method still worked. Maintaining a constant temperature (37 °C) during expression simplifies the experimental pipeline. Furthermore, it prevents induction of the host cells' heat shock response, which could potentially result in increased expression of undesired proteins such as proteases[54]. Utilizing an expression host with a better matching codon usage could potentially increase the protein synthesis rate and thereby result in higher methyltransferase activity[55]. Chemical gene

synthesis of the methyltransferases would allow for individual codon optimization, which may also help to increase the translation of the methyltransferases.

In the present study, we were able to analyze a pool of up to 19 linearized plasmids in a single SMRT cell while reaching average coverage of 4750-fold. This indicates that even a greater number of methyltransferases could potentially be analyzed in one sequencing cell. However, some methyltransferases stand out by having low coverage e.g. the negative control with the *M. thermoacetica* motif cassette and different modification types require different coverage. Therefore, increasing the number of plasmids pooled for sequencing beyond 20 is not recommended if high detection rates are anticipated. By using larger sized chips the number of plasmids pooled could be further increased. The chip on the Sequel System of PacBio is predicted to handle a pool of 200–300 plasmids, because of improved read-length and loading. In this study m5C methylation was detected by PacBio, but with low accuracy despite a coverage higher than 250-fold. Improvement to accuracy and sensitivity when detecting m5C modifications can be achieved by oxidization of the modified cytosine[56] or by utilizing another sequencing technology[26–28].

The application of SMRT and nanopore sequencing has empowered the identification of DNA modifications in a much faster manner than before[57]. This has facilitated better understanding of methylation in cell physiology (mostly based upon sequencing of entire genomes)[29]. However, detecting patterns of single methyltransferases remain a challenging task. Heterologous expression of methyltransferases (from plasmid) and subsequently assessing the methylation patterns by SMRT sequencing of both plasmids (various plasmids are needed as methylation sites needs to be present) and genomes of the expressing hosts[2,39,58] has been the method of choice. Modification by all three major types of methyltransferases (Type I, II, and III) has been determined, and clearly demonstrates the applicability of SMRT sequencing to decipher the recognition sites of methyltransferases. Compared to the previous time-consuming methods based upon restriction digestion[6] the gain has been massive. The presented method shares main principles of determining modified bases with the previous studies, but targets a more streamlined high throughput approach for both cloning and analysis. This is facilitated by using the same expression vector with a strain specific motif cassette for the methyltransferases, enabling identification of motifs solely based upon the cassette. This was demonstrated by including Type I methyltransferases (two target sequences separated by typical 4–8 nucleotides), which requires a large sequence space to obtain the correct motifs. The two mesophilic Type I methyltransferases were assigned with high specificity to motifs with modQV above 1,000, whereas the thermophilic had slightly extended motifs and lower ModQV (causes are discussed above), but still possible to identify the activity and correlate it to the methylome. Methyltransferases with no activity detected could potentially be functional but not active on the genome. Such sites could potentially be further analyzed by PacBio sequencing the genome of the *E. coli* strains expressing the individual methyltransferases, thereby obtaining higher specificity in the motif-recognition.

Furuta et al.[40] assigned specificity to methyltransferases or to their specificity-determining domains (TRD) by assessing different strains of *Helicobacter pylori* for differences in the repertoire of methylation motifs and in specificity-determining genes. A commensurate approach of sequencing closely related strains was done by Chen et al.[35], additionally they combined it with prediction of sequence specificity using SEQWARE[59]. Such a combinatorial approach holds great potential, but sample space has some requirement as it need a specific diversity assuring differences in methylome along with minor changes in the genomes

(preferable only in the TRD). The developed method does not have any requirements for a specific diversity on methylome or DNA level. It is anticipated that the data generated with the presented method could contribute in increasing the specificity of the combinatorial methods.

The two strains *M. thermoacetica* and *A. woodii* had both substantially modified genomes. Using the developed method MetMap, we were able to couple the genetic sequences to the methylome patterns in a high-throughput manner. Of the 12 motifs identified by PacBio sequencing of the genome, we were able to directly assign 11 methyltransferases, corresponding to an efficiency of 92%. With the currently available predictive tools (REBASE) nine of the evaluated methyltransferases had assigned specificity, however only in four cases (all from *A. woodii*) were there a correlation with the specificity detected in this study. By analyzing larger numbers of methyltransferases, the experimentally validated knowledge gained could undoubtedly improve the accuracy of algorithms predicting protein-DNA binding sites based on the protein coding sequence.

## Methods

**Strains and cultivation**. The strains and plasmids used in this study are listed in Supplementary Data 2 and the DNA oligomers used for the constructs are listed in Supplementary Data 3. *E. coli* Top10 (Thermo Fisher Scientific) was used for plasmid constructions, whereas *E. coli* Stellar™ dcm−/dam− (Clonetech) was used for expressing the methyltransferases. Both *E. coli* strains were routinely cultivated in LB (Luria-Bertani) broth at 37 °C with shaking. For plasmid propagation TB (Terrific Broth) was used[60,61]. When required respective antibiotic was supplemented to the media. *M. thermoacetica* ATCC 39073-HH was grown at 60 °C in un-defined media (Supplementary Table 2, in accordance with Daniel et al.[62]) utilizing fructose as carbon source. *A. woodii* DSMZ 1030 was grown in DSMZ medium 135 strictly anaerobically at 30 °C.

**Genome extraction**. Cultures of *M. thermoacetica* and *A. woodii* were grown until exponential growth was reached. Then gDNA was extracted by Wizard® Genomic DNA Purification Kit (Promega), according to manufactures guidelines, with the following modifications. The lysis step with 10 mg/ml lysozyme was extended to 2 h and gDNA was resuspended in 1 mM Tris-HCL (pH 7.4) and 0.1 mM EDTA (pH 8.0). DNA quantification was done fluorometrically utilizing a Qubit™ instrument (Thermo Fisher Scientific).

**Genome sequencing**. *M. thermoacetica* was sequenced using a PacBio RSII instrument (Pacific Biosciences). DNA was sheared into 10 kb fragments using a Genemachines HydroShear Instrument (Digilab). SMRT-bells were constructed according to the manufacturer's instructions (Pacific Biosciences). SMRT-bells were sequenced on two SMRT-cells on a Pacific Biosciences RSII sequencer according to the manufacturer's instructions with 4 h movie-time. Only reads longer than 10 kb were considered and min QV of 85 considered. The average reference coverage was above ×137, from 49,792 reads, with an average read length of ~11,000 bp.

**Program for designing motif cassettes**. A script was designed to randomly organize the motifs initially identified by the PacBio sequencing of the genomes and stitch them together in adequate number, resulting in the strain-specific motif cassette. Furthermore, the program separates each motif by a 'n'. The user submits the motifs to the program, classified by two rules. Rule 1: motifs of 2–10 n´s separated by two specific sites, typical for Type I methyltransferases. Rule 2: motifs not having separated specificity sites, generally for Type II and III. This differentiation is included to assure higher number of motifs for Type I motifs (12 vs. 10 motifs). For motifs with ambiguous nuclotides, are de-ambigulated, according to two rules: Rule 1, pick L random variants of the motif. As an example, the motif ATGNNTTA has a total of 16 possible actual sequences. If $L < 16$ then the program will random pick L variants (without duplicates). If $L > 16$ then each possible variant will be picked at least L/16 times and some will be picked 1 more than that. Rule 2, make K copies of each completely "de-ambiguated" variant: E.g. the sequence "SATC" will then be treated as two sequences: "GATC" and "CTAC" that will each appear in K copies. The program outputs optional number of motif cassettes as individual genbank and fasta files. The script for designing the motif cassettes was developed to run in python 3.6 or newer and can be found together with program and installation descriptions at https://github.com/biosustain/metmap.

**Construction of plasmids**. Plasmids (listed in Supplementary Data 2) were constructed by PCR-based uracil-specific excision reagent (USER) cloning method[63,64] or using Gibson cloning[65]. For USER cloning, one microliter of 5x HF buffer

(Thermo Scientific) and 1U of USER™ enzyme mix (New England Biolabs, 1 U/μl) were added to 8 μl of the mixture of purified PCR products, plasmid-backbone or genes. All PCR products were amplified with oligonucleotides (Integrated DNA Technologies) having uracil incorporated, utilizing Phusion U polymerase (Thermo Fisher Scientific). The reaction mixture was incubated for 25 min at 37 °C, followed by 25 min of incubation at a temperature optimized for annealing of the fragments for 25 min. Eight microliters MilliQ was added to the reactions, reaching a final volume of 20 μl. In all, 2.5 μl of the diluted USER reaction was used to transform 50 μl competent *E. coli*. Gibson assemblies were carried out using Gibson Assembly Cloning Kit (NEB). Fragments (motif cassette) with ~30 bp overlap regions were mixed together with the backbone fragment in 1:1 molar ratio together with an equal volume of Gibson master mix, incubated at 50 °C for 1 h and immediately transformed into *E. coli*.

**Automated cloning**. An automated cloning workflow was set up on a Hamilton Microlab VANTAGE Liquid Handling System (Hamilton Bonaduz AG). The system was fully customized to accommodate complex strain construction work-flows. It was equipped with a modular arm holding eight single pipetting channels (Single Channel 1000 μL NanoPulse), a 96-channel pipetting head (MPH, Mul-tiprobe Head), a camera, a barcode reader (ID Loading Device (1D & 2D Barcode Reading)), four heater shakers (Hamilton), a vacuum station CVS (Hamilton), a HEPA hood (Hamilton), a Microplate centrifuge and automated Microplate cen-trifuge loader (Agilent Technologies Inc), and a TRobot thermocycler (Analytik Jena AG). Cloning was based on USER cloning using general conditions described above. Analysis of DNA fragments by high throughput capillary electrophoresis was performed off-deck on a Caliper LabChip GXII (PerkinElmer Inc.).

The liquid handling system was loaded with genomic DNA from the different organisms, oligonucleotides (Supplementary Data 3) for amplifying the 23 methyltransferases and a negative control (the *bgaB* gene from *Geobacillus stearothermophilus*, refseq WP_020755758.1), and plasmid backbone holding a strain-specific motif cassette. Clones were verified by colony PCR and subsequent Sanger sequencing of the insert. To reach complete cloning efficiency, the automated cloning was complemented by manually cloning, allowing for individual optimization of selected PCR conditions.

**Plasmid extraction**. Fresh medium (supplemented with antibiotic) was inoculated with 10% overnight culture. Growth was re-established by incubation at 37 °C for 1 h before induction with IPTG (final concentration of 1 mM). The induced cultures were incubated for additional 3 h. The plasmids were purified using a NucleoSpin® Plasmid (Macherey-Nagel) spin column kit. DNA was quantified fluorometriccally utilizing Qubit™ (Thermo Fisher Scientific). Prior to pooling, the plasmids were linearized with NotI (Thermo Fisher Scientific) and the enzyme was inactivated by incubation at 80 °C for 10 min.

**SMRT sequencing of pooled plasmids**. A SMRT-bell library was constructed for each pool of linearized plasmids according to the manufacturer's instructions (Pacific Biosciences). SMRT-bells were sequenced on one SMRT-cell per pool on a Pacific Biosciences RSII sequencer according to the manufacturer's instructions with 4 h movie-time.

**Bioinformatics/sequence handling**. Analysis of the methylation profile and motif detection was performed in SMRT-Portal v.2.3.0 as follows: Reads for the indivi-dual methyltransferase sequences and negative controls were separated in two steps: an initial sorting based on the motif cassette was followed by grouping based on the specific methylase sequence. The methylation status of the selected sequencing reads was then determined using the 'Modification and Motif_Analysis' pipeline included in SMRT-Portal (details are available at https://www.pacb.com/wp-content/uploads/2015/09/SMRT-Pipe-Reference-Guide.pdf). When motif identification by standard parameters was too ambiguous and or appeared exten-ded, optimization was conduced to assure better motif identification. In particular, measures were taken to reduce coverage into the by the manufactures reported coverage range, including adjustment of min read length, readQV, and ModQV.

**Reporting summary**. Further information on research design is available in the Nature Research Reporting Summary linked to this article.

## Data availability

The datasets generated and analyzed during the current study are available online, including the newly annotated genome of *M. thermoacetica* 39073-HH (Genbank accession number CP031054) and SMRT sequencing files SRR8358415, and SRR8357235 Motif_summary.csv files are supplied as Supplementary Data 1.

## Code availability

The developed script for designing the motif cassettes can be found together with program and installation description at gibhub [https://github.com/biosustain/metmap].

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

## Acknowledgements

We would like to acknowledge support of the National Genomics Infrastructure (NGI)/Uppsala Genome Center and UPPMAX for providing assistance in massive parallel sequencing and computational infrastructure. The annotation of the genome of *M. thermoacetica* 39073-HH was done by Christian Bille Jendresen, his efforts were greatly appreciated. This study was funded by European Union's Horizon 2020 LCE, "AMBI-TION"; The Novo Nordisk Foundation; and the Independent Research Fund Denmark. Work performed at NGI / Uppsala Genome Center has been funded by RFI / VR and Science for Life Laboratory, Sweden. The project was partly funded by the ERA-IB project "CO2CHEM" (ERA-IB-14-038).

## Author contributions

A.T.N. supervised the project. T.Ø.J. designed the experiment. The cloning work, both manually and automated, were carried by T.Ø.J., S.R., J.M. and S.A.B.J. C.T-R. carried out the PacBio sequencing and respective data handling. L.E.P developed the program for designing the motif cassettes. T.Ø.J., C.T.-R. and A.T.N. analyzed the data and polished the paper. All authors discussed the results and assisted during paper preparation.

## Additional information

**Competing interests:** The authors declare no competing interests.

