## [Peer Review File · Nature Communications]

Reviewers' comments:

Reviewer #1 (Remarks to the Author):

Review of MedMap... by Jensen et al. for Nature Communications.

After Pacbio decoding of a bacterial methylome, it is necessary but often difficult to relate each methylation motif to a methyl transferase gene (more accurately, Target Recognition Domain). In order to address this important issue, there have been several experimental works and also bioinformatic works on the evolution of these genes and their targets for several years, especially on the nature of target recognition domains and their mobility (eg. doi.org/10.1371/journal.pgen.1004272 and related papers).

The authors modifies the method used earlier. This trick takes advantage of the long-reading capacity and methylation-decoding ability of Pacbio. Instead of one methylation motif, the plasmid for cloning and expression of MTase now carries a cluster of all the motifs from a strain. The authors claim a high-throughput method, but the cloning of the methyltransferase genes is as usual. After induction of each MTase separately, each plasmid is recovered. Their mixture is decoded by Pacbio to decode separately the methylation status of the cluster linked to a MTase. This method is called high-throughput because it starts with a mixture of plasmids with different MTase genes. However, the individual decoding did not work very well. Therefore, a two-step procedure was necessary.

The problem likely comes from competition between different templates for occasion of replication. Size selection of genomic DNA in cloning or isolating fragments of a similar size may bring in equality in replication. A cleaner result will be obtained if a fragment of similar size (<1 kb) containing the motif cluster and a small part from the cloned MTase gene is connected to hairpins and read many times. The hairpins need not contain barcodes for multiplexing but the barcodes may help to find out the problem.

The present results do not show the best performance of Pacbio now. The barcoding method works better with Pacbio Sequel than RSII. It will be able to decode 300 Gb/ cell in 2019, which will accommodate 3×10^7 plasmids.

Another problem is that sequence specificity is a function of the expression level of MTase. The ambiguity in specificity might be their essence. Refer to recent papers on their evolution.

Two methylomes are enough to claim a method paper? $2^3 \times 1/2$ is worth the name of high throughput?

In a bacterial strain with a natural transformation system, this method can be combined with selection of MTase cloning by protection of the site on the plasmid from restriction. The methylation motif can be systematically randomized to represent one of the many. Then Pacbio-decode the mixture of the plasmids.

Overall, the sentences can be more accurate and concise as scientific statements. Two examples. The most (1st line in Abstract): Show convincing evidence or delete. Many similar expressions. Figure 1. Figure showing the workflow of the described method named MetMap.
> Figure 1. MetMap workflow.

The manuscript can be reduced to 1/3 ~ 1/4 for clarity.

Figure 1. Fonts can be hardly read in Win or Mac. Most letters are unnecessary for clear presentation, though.

Figures 1, 2 and 3 can be combined.

Table 4,5. Better presentation? To supplements?

32. base modification

38. phase variation > global gene expression regulation

49-51. Be specific.

53. More papers.

86. 5mC. Sensitivity and selectivity of detection in your system?

89. Is CCWGG methylation detected in the other strain? Cut the genomes with EcoRII.

121-123. Mention Specificity subunit or TRD here or in Introduction.

125-131. Separately in one step is simple and straightforward and not very low throughput (see above).

Discussion. Too lengthy and empty. Esp. 278-282. Discuss recent papers on sequence recognition by RMs, their mechanisms and evolution.

257-. Unclear.

259. 133. Cite evidence.

262-274. Not convincing.

275-. Relevant? Delete?

References. Not up to date at all.

Ref. 1. More general than m6A?

1,2,4,5,7. More recent reviews?

8. Newer reference.

REBASE--a database for DNA restriction and modification: enzymes, genes and genomes. Roberts RJ, Vincze T, Posfai J, Macelis D. Nucleic Acids Res. 2015 Jan;43(Database issue):D298-9. doi: 10.1093/nar/gku1046.

Reviewer #2 (Remarks to the Author):

Review of Jensen et al.

This is a very cute method that addresses an important problem in assigning recognition specificities to specific methylases. As the authors note, experimental confirmation is an important step in assigning function with certainty. The experimental work is very solid and the results are

excellent. My main concern stems from the write up, which really should be improved to provide guidance to others wishing to use this method themselves. There are also some points related to previous work by others that should be discussed within the context of this new method. I apologize for the extensive number of comments, but I like the method very much and hope that others will use it. My comments are merely aimed at improving the usefulness of the paper when it is finally published.

1. The starting point for designing the test oligonucleotides is the interpretation of the PacBio motif analysis. The authors should be explicit in noting that the predicted motifs are sometimes less than fully accurate and some interpretation is needed. For instance, of the motifs predicted for *M. thermoacetica* a number are clearly miscalls (e.g. TNBNNTBHNH is clearly incorrect) and should not be built into the test oligonucleotide. This point should be made in the Methods section and some indication given by the author of how decisions about the credibility of the motifs is determined, prior to using them. Since this point comes up several times, it would be worth dealing with that at the start of the Results since it plays directly into cassette design.

2. The *A. woodii* sequence and motifs had been determined previously, whereas the *M. thermoacetica* sequence is new. Was *A. woodii* newly sequenced or were the earlier sequences and motifs used? This should be stated explicitly, since the Type I recognition sequence is different from that reported in Blow et al., which was also determined by PacBio sequencing. In particular, the newly reported sequence is less specific than that reported previously. Were all variations of the 5'-submotif tested (i.e. GATA, GATC, GATG and GATT)? How certain can the authors be that this is now the correct recognition sequence for this Type I enzyme? This needs some more explanation in the Results given that only 12 target sequences were present in this study, whereas in the original genome there are many more sites that were queried.

3. I was unable to find the sequences of the composite oligonucleotides that contained the various motifs for the two strains. These must be included in the supplementary material with the various motifs obtained from the PacBio data on the genome clearly indicated.

4. Line 93. "without experimental evidence". This should be modified since the presence of the PacBio methylation signal does constitute experimental evidence.

5. Lines 111-112. At some point it should be mentioned that if desired the complete genome of the *E. coli* carrying the plasmid could be sequenced too. This could be useful in resolving differences between plasmid-determined motifs and ones initially present in the original genome.

6. Line 133. Looking at Tables 2 and 3, I count 19 motifs, although not all are believable. However, I couldn't easily reduce it to 15. Perhaps, this should be rephrased so as not to be misleading.

7. Line 135-143. This point is a very important one that should be introduced earlier (see comment #1). The interpretation of the PacBio motifs generated by the PacBio SMRT Analysis often requires human analysis as it does here. This applies to both the initial definition of motifs and the subsequent analysis of the results obtained by this method.

8. Line 160. It was unclear to me what was meant by "showed activity in both directions". Do the authors mean both strands were modified?

9. Line 166-167. The motifs shown in the motif_summary.csv files illustrate well the problems that arise with too much or too little coverage. In both cases the motif is incorrectly called and the interpretation that the authors settle on is not conclusive. If they want to argue for the reduced length motif, they should present their reasoning.

10. In Table 2, the Gene names in column 1 don't match the gene names in the GenBank file CP031054. This is very confusing since the gene names in the GenBank file seem closer to those in the current *M. thermoacetica* file in GenBank (CP000232). Is the submitted version of the GenBank file not the final one? This needs to be fixed. Also, when the final version of the GenBank file is deposited it would be helpful to include the motif_summary.csv files for this genome.

11. In Table 2, the motif for the Type I system is very odd. It is something that is commonly seen when the coverage is very high (524). In this case because there is only one Type I motif and only one candidate in the original genome there is no problem in proposing that this can be interpreted, but if more than one Type I system was present this would be problematic. This should be mentioned in the Results.

12. Line 176. The CTCCG motif found in the original genome appears genuine and the best

explanation for not detecting it in the test set is that all of the genes necessary to produce activate methylation are not present in the set. One possible explanation could be that the two separate methylase gene fragments (2238 and 2241) are in fact active if both are expressed or alternatively a third gene is necessary to activate 2241. In any event this is a cautionary note that should be mentioned. It might even be worth repeating the experiment with both ORFs included in the plasmid. If that turns out to be the explanation, then this is a point that can be made very solidly within the paper.

13. Line 178-180. The motif_summary.csv file available to this reviewer showed only 3 motifs that were completely uninterpretable. If ORF 18590 really is the m5C methylase CCWGG then this result is not surprising since contrary to the earlier statement (line 140) 250 fold coverage does not guarantee correct identification of an m5C motif. This result could be used to emphasize this problem, which is not well appreciated generally, and apparently not by these authors.

14. Line 185. Two of these methylase genes (34920 and 35180) are encoded on prophages, which might account for their apparent inactivity. They may need additional genes to activate them or their recognition sequences may not be present since prophage methylases are not usually expressed in the genome and so no motifs would be present in the original genome analysis.

15. Lines 233-249. Having attempted to clone many genes from thermophilic organisms into a mesophile such as *E. coli*, I would be a little more cautious in claiming that they are usually expressed with little difficulty. In some cases additional genes are necessary for activity.

16. Line 254. This requirement for 250-fold coverage for m5C detection should not be repeated here. PacBio is just not suited to m5C detection, although occasionally it can give some indication of an active m5C methylase. There are other better methods for this.

17. Line 262. While it is true that Clark et al. only included Type II and III methylases, a later paper (ref 23 in this manuscript) did include a number of Type I systems.

18. Line 272-274. The Clark et al. paper examined the complete bacterial genomes in which the methylase genes were cloned, not just the plasmids. This provided a very large sequence space in which the motifs could occur.

19. Lines 287-289. While this is true and I have no problem with it being included, I would note that we are going to stop making predictions as flimsy as some of the current ones now that we are establishing a gold standard set of methylases with experimentally verified recognition sequences. The results from this paper help in that endeavor and I hope others will decide to use it and generate yet more experimental data.

Some of the points raised above could usefully be included in the Discussion (or expanded in the Results), since it is always helpful to the user to know why apparent failures might show up.

Rich Roberts

Reviewers' comments:

Reviewer #1 (Remarks to the Author):

Review of MedMap... by Jensen et al. for Nature Communications.

After Pacbio decoding of a bacterial methylome, it is necessary but often difficult to relate each methylation motif to a methyl transferase gene (more accurately, Target Recognition Domain). In order to address this important issue, there have been several experimental works and also bioinformatic works on the evolution of these genes and their targets for several years, especially on the nature of target recognition domains and their mobility (eg. doi.org/10.1371/journal.pgen.1004272 and related papers).

Throughout the revised manuscript we have emphasized more specific formulation with respect to TRD. We have included the suggested study as well as similar ones, in order to make the reflections and discussions of the developed method broader.

Example from the introduction: "diversity of methylation between species and strains as well as mobility of RM systems or target recognition domain (TRD) has become apparent^{32,33}. Approaches comparing methylomes and genomes to couple TRD of methyltransferases to specific patterns have been published, enabled by a specific degree of diversity and/or size of the compared groups^{34,35}."

The authors modifies the method used earlier. This trick takes advantage of the long-reading capacity and methylation-decoding ability of Pacbio. Instead of one methylation motif, the plasmid for cloning and expression of MTase now carries a cluster of all the motifs from a strain. The authors claim a high-throughput method, but the cloning of the methyltransferase genes is as usual. After induction of each MTase separately, each plasmid is recovered. Their mixture is decoded by Pacbio to decode separately the methylation status of the cluster linked to a MTase. This method is called high-throughput because it starts with a mixture of plasmids with different MTase genes. However, the individual decoding did not work very well. Therefore, a two-step procedure was necessary.

Thanks a lot for your comment. Based on the streamlined pipeline, and use of automation both in the cloning and in the design phase, we consider the methods to be high-throughput and scalable.

The two-step sorting was included to allow sorting of methyltransferases from closely related strains (potentially having only a few differences in the methyltransferases genes). If this sorting was not included, very close related methyltransferases could be grouped together, which would be undesired. In the published work on mobility of TRD, it has been reported that few mutations in TRD can cause change in specificity of the methyltransferases. This was accounted for by the two step sorting.

The problem likely comes from competition between different templates for occasion of replication. Size selection of genomic DNA in cloning or isolating fragments of a similar size may bring in equality in replication. A cleaner result will be obtained if a fragment of similar size (<1 kb) containing the motif cluster and a small part from the cloned MTase gene is connected to hairpins and read many times. The hairpins need not contain barcodes for multiplexing but the barcodes may help to find out the problem.

Thanks for your thought, but we do not believe that the use of shorter fragments will solve the problem. If two closely related organisms are analyzed, the orthologous genes will most often have a too high sequence identity and cannot be separated by simple mapping.

The present results do not show the best performance of Pacbio now. The barcoding method works better with Pacbio Sequel than RSII. It will be able to decode 300 Gb/ cell in 2019, which will accommodate 3×10^7 plasmids.

The reason for not using barcoding was to save time and money. Obviously barcoding would enable us to isolate individual sequences, however it would take time in the preparation and it is costly, which was why this option was excluded.

Another problem is that sequence specificity is a function of the expression level of MTase. The ambiguity in specificity might be their essence. Refer to recent papers on their evolution.

Thanks for your comment. We have now included more recent papers on evolution. We acknowledge that expression levels (or reduced activity by heterologous expressed proteins) will influence to ambiguity in the methyltransferases specificity. In the developed method we couple the methylome data from sequencing of gDNA with the motifs found by individual expression. Thereby the influence by different expression levels should be minor. Furthermore, if ambiguous motifs are identified in the methylome, it is included in the motif cassette thereby it will be possible to distinguish.

Two methylomes are enough to claim a method paper? $23 \times 1/2$ is worth the name of high throughput?

We have developed a method, which aims at being streamlined and high throughput in both data generation and data evaluation. In our opinion, the number of methyltransferases included does not determine whether a method is high throughput or not. It is rather the concept and the speed by which data can be generated.

In a bacterial strain with a natural transformation system, this method can be combined with selection of MTase cloning by protection of the site on the plasmid from restriction. The methylation motif can be systematically randomized to represent one of the many. Then Pacbio-decode the mixture of the plasmids.

We acknowledge the suggestion for further develop of the method, however found it outside the scope of this manuscript.

Overall, the sentences can be more accurate and concise as scientific statements. Two examples.

The most (1st line in Abstract): Show convincing evidence or delete. Many similar expressions.

Figure 1. Figure showing the workflow of the described method named MetMap.

> Figure 1. MetMap workflow.

The manuscript can be reduced to $1/3 \sim 1/4$ for clarity.

The manuscript has been revised to make the language more accurate and concise. Paragraphs have deleted to reduce the size of the manuscript and to increase clarity.

Figure 1. Fonts can be hardly read in Win or Mac. Most letters are unnecessary for clear presentation, though.

We acknowledge the issue with the compiled manuscript. In the high-resolution pictures available online from journal the fonts is readable.

Figures 1, 2 and 3 can be combined.

Thanks a lot of the suggestion. Fig 1 and 2 has now been combined. For clarity fig 3 was kept separately.

Table 4,5. Better presentation? To supplements?

The tables have been moved to supplementary.

32. base modification

Correction made

38. phase variation > global gene expression regulation

Correction made

49-51. Be specific.

Technologies for detecting methylation have now been specified

“Recently, genome-wide analyses of methylomes have been published, relying on Single Molecule Real-Time DNA sequencing (SMRT) for direct detection of methylated DNA nucleotides²⁵, thereby making it possible to recognize all three types of bacterial methylation². Other sequencing methods, including bisulfite sequencing^{26,27}, Nanopore technology, also have the potential to detect modifications of DNA^{28,29}. “

53. More papers.

More papers have been included, such as:

Rao, D.N. et al. Methylome diversification through changes in DNA methyltransferase sequence specificity. PLoS Genet. 10, e1004272 (2014).

Furuta, Y., Kawai, M., Uchiyama, I. & Kobayashi, I. Domain movement within a gene: a novel evolutionary mechanism for protein diversification. PLoS One 6, e18819 (2011).

86. 5mC. Sensitivity and selectivity of detection in your system

In this study 5mC methylation was detected by PacBio, although with low accuracy despite a coverage higher than 250-fold. 5mC detection is considered to be a general problem with PacBio. In the revised manuscript we have been explicit in our discussion of 5mC detection. However, it must be noted that the developed method also can utilize other sequencing methods that are capable of detecting methylation.

89. Is CCWGG methylation detected in the other strain? Cut the genomes with EcoRII.

The methylation-motif was not detected in M. thermoacetica. gDNA of M. thermoacetica are digested with EcoRII, whereas gDNA of A woodii are not.

121-123. Mention Specificity subunit or TRD here or in Introduction.

In the introduction it is stated that difference in the TRD or specificity subunit, will alter the motif recognition.

125-131. Separately in one step is simple and straightforward and not very low throughput (see above).

We are unfortunately unsure what is meant with this comment. If the reviewer could please elaborate, then we will be happy to address it.

Discussion. Too lengthy and empty. Esp. 278-282. Discuss recent papers on sequence recognition by RMs, their mechanisms and evolution.

Thanks for your relevant comment. Parts of the discussion have now been deleted and additional methods have been discussed, to assure that the discussion is more "to the point". We do believe that this has focused the discussion significantly.

257-. Unclear.

The sentence has been deleted.

259. 133. Cite evidence.

It is estimates based on the improved read-length and loading of the Sequel System which was recalculated. This part has been rewritten to make it more explicit.

"The chip on the Sequel System of PacBio is predicted to handle a pool of 200-300 plasmids, because of improved read-length and loading."

262-274. Not convincing.

This section has been deleted, as the discussion has been rewritten.

275-. Relevant? Delete?

This has been deleted in the revised discussion.

References. Not up to date at all.

More recent publication have been included.

Ref. 1. More general than m6A?

More general papers have been included.

1,2,4,5,7. More recent reviews?

More recent reviews have been included.

8. Newer reference.

REBASE--a database for DNA restriction and modification: enzymes, genes and genomes.

Roberts RJ, Vincze T, Posfai J, Macelis D. Nucleic Acids Res. 2015 Jan;43(Database issue):D298-9. doi: 10.1093/nar/gku1046.

This reference has been included.

Reviewer #2 (Remarks to the Author):

Review of Jensen et al.

This is a very cute method that addresses an important problem in assigning recognition specificities to specific methylases. As the authors note, experimental confirmation is an important step in assigning function with certainty. The experimental work is very solid and the results are excellent. My main concern stems from the write up, which really should be improved to provide guidance to others wishing to use this method themselves. There are also some points related to previous work by others that should be discussed within the context of this new method. I apologize for the extensive number of comments, but I like the method very much and hope that others will use it. My comments are merely aimed at improving the usefulness of the paper when it is finally published.

1. The starting point for designing the test oligonucleotides is the interpretation of the PacBio motif analysis. The authors should be explicit in noting that the predicted motifs are sometimes less than fully accurate and some interpretation is needed. For instance, of the motifs predicted for *M. thermoacetica* a number are clearly miscalls (e.g. TNBNNTBHNNH is clearly incorrect) and should not be built into the test oligonucleotide. This point should be made in the Methods section and some indication given by the author of how decisions about the credibility of the motifs is determined, prior to using them. Since this point comes up several times, it would be worth dealing with that at the start of the Results since it plays directly into cassette design.

First of all, thanks a lot for your encouraging and positive comments. There was a miscorrelation between the motif_summary.csv and the data presented in table 1. Unfortunately, the motif_summary.csv file was derived from the first analysis of the M. thermoacetica and changed quite significantly after a software update. Now the correct motifs_summary file has been attached (and attached to the genome). To assure transparency in our evaluation of the modifications, a table that lists all data from the motif_summary.csv files has now been included as supplementary material. In the text it has now been emphasized that human evaluation are involved for interpretation of PacBio sequencing/modification results. The criteria in which we have chosen motifs have also been elaborated in the methods.

2. The *A. woodii* sequence and motifs had been determined previously, whereas the *M. thermoacetica* sequence is new. Was *A. woodii* newly sequenced or were the earlier sequences and motifs used? This should be stated explicitly, since the Type I recognition sequence is different from that reported in Blow et al., which was also determined by PacBio sequencing. In particular, the newly reported sequence is less specific than that reported previously. Were all variations of the 5'-submotif tested (i.e. GATA, GATC, GATG and GATT)? How certain can the authors be that this is now the correct recognition sequence for this Type I enzyme? This needs some more explanation in the Results given that only 12 target sequences were present in this study, whereas in the original genome there are many more sites that were queried.

We did not sequence the A. woodii strains, this has been clarified. Data has been cross-checked to assure that the data generated by Blow et al. was used for constructing the motif cassette. With respect to the variation in the 5' submotif, GATG was included in the motif cassette. Why the 5' submotif was identified as GATN is unknown, we hypothesis that it is caused by the "relative" small number of motifs in the cassette. Since, the variance in the

genome is greater the motif, that motif would be correct. This has been elaborated in the text.

From the result section: "Motif identification of Awo_c17130_Awo_c17140 (at QV255) resulted in two motifs that appeared as complement, despite diminutive differences. Compared to the motif found in the methylome of A. woodii³⁰ an ambiguous nucleotide was present on the fourth position of the 5'-submotif, thus it is reasonable to link activity of the methyltransferases to motifs observed in the genomes."

3. I was unable to find the sequences of the composite oligonucleotides that contained the various motifs for the two strains. These must be included in the supplementary material with the various motifs obtained from the PacBio data on the genome clearly indicated.

Thanks for pointing this out. The sequences have been included as Supplementary material (figure S4 and S5). It must be noted that the MetMap-script for generating the motif cassettes is randomizing the motifs, therefore it will not be likely to "recreate" the exact same motif cassette with the program.

4. Line 93. "without experimental evidence". This should be modified since the presence of the PacBio methylation signal does constitute experimental evidence.

It has been deleted

5. Lines 111-112. At some point it should be mentioned that if desired the complete genome of the E. coli carrying the plasmid could be sequenced too. This could be useful in resolving differences between plasmid-determined motifs and ones initially present in the original genome.

This is a very good point - a comment has been added in the discussion.

From Discussion: "Methyltransferases with no activity detected could potential be functional but not active on the genome. Such sites could potentially be further analyzed by PacBio sequencing the genome of the E. coli strains expressing the individual methyltransferases, thereby obtaining higher specificity in the motif-recognition"

6. Line 133. Looking at Tables 2 and 3, I count 19 motifs, although not all are believable. However, I couldn't easily reduce it to 15. Perhaps, this should be rephrased so as not to be misleading.

As mentioned above, all modification data have been included as Supplementary material. Furthermore, the paragraph has been rephrased to assure a more accurate formulation.

From results: "Of the 23 methyltransferases and two negative controls, it was possible to isolate all coding sequences and identify 21 modification motifs with a detection rate close to 100% for all (table 2 and 3). Additional motifs were detected (Supplementary table S3 lists all information from the motif_summary.csv files) under the given parameters, however upon inspection some appeared as miscalled motifs."

7. Line 135 143. This point is a very important one that should be introduced earlier (see comment #1). The interpretation of the PacBio motifs generated by the PacBio SMRT Analysis often requires human analysis as it does here. This applies to both the initial definition of motifs and the subsequent analysis of the results obtained by this method.

*We agree that interpretation of PacBio generated data often is required. However, when assessing the methylome data (of *M. thermoacetica*), the data listed in table 1 was generated by the parameters described in the methods section, no further interpretation was utilized. This should be apparent with the correct motif_summary.csv and modification report attached. Under the initial definition of the motifs miscalled nucleotides was addressed.*

The requirement of human analysis of PacBio motifs has now additionally been mentioned in the introduction, and the description in the methods section has additionally been elaborated.

8. Line 160. It was unclear to me what was meant by “showed activity in both directions”. Do the authors mean both strands were modified?

Thanks for pointing this out. We have now changed the phrasing to “activity on both strands”.

From results: “Two sets (Awo_c08800_Awo_c08810 and Awo_c17130_Awo_c17140) demonstrated activity on both strands,…”

9. Line 166-167. The motifs shown in the motif_summary.csv files illustrate well the problems that arise with too much or too little coverage. In both cases the motif is incorrectly called and the interpretation that the authors settle on is not conclusive. If they want to argue for the reduced length motif, they should present their reasoning.

Very relevant comment, we acknowledge the problem that arise when coverage becomes either too high or too low. Both cases will lead to misinterpretation of the data. In the developed method we use the methylome as reference, thereby having indications the motifs we wish to find when heterologous expressing the methyltransferases. Throughout the revised manuscript we have emphasized more on the human analysis/interpretation of the PacBio results. A paragraph has been included in the methods descriptions on how sequencing are analysis is optimized to obtain better reads.

*From results: “ Motif identification of Awo_c17130_Awo_c17140 (at QV255) resulted in two motifs that appeared as complement, despite diminutive differences. Compared to the motif found in the methylome of *A. woodii*³⁰ an ambiguous nucleotide was present on the fourth position of the 5´-submotif, thus it is reasonable to link activity of the methyltransferases to motifs observed in the genomes.”*

10. In Table 2, the Gene names in column 1 don't match the gene names in the GenBank file CP031054. This is very confusing since the gene names in the GenBank file seem closer to those in the current *M. thermoacetica* file in GenBank (CP000232). Is the submitted version of the GenBank file not the final one? This needs to be fixed. Also, when the final version of the GenBank file is deposited it would be helpful to include the motif_summary.csv files for this genome.

Thanks for catching this. There was a mismatch between the genes-names of CP031054 and the naming in the first version of the manuscript. This was because the naming used in the manuscript derived from a premature version of the genome. In the revised manuscript the gene-names have all been changed to assure coherence between the data presented and the genome uploaded.

11. In Table 2, the motif for the Type I system is very odd. It is something that is commonly seen when the coverage is very high (524). In this case because there is only one Type I motif and only one candidate in the original genome there is no problem in proposing that this can

be interpreted, but if more than one Type I system was present this would be problematic. This should be mentioned in the Results.

Despite the extended motif, it was clear that the motif correlates to the motifs in the methylome. However, this is a point of uncertainty, which has been further elaborated in the results section and further discussed in the discussing section.

From Results: "When heterologous expressed, two extended and complement motifs were detected, which could be linked to the motif ATCNNNNNCTC identified on the genome (if more Type I motifs were present in the genome this may be more challenging)."

12. Line 176. The CTCCG motif found in the original genome appears genuine and the best explanation for not detecting it in the test set is that all of the genes necessary to produce activate methylation are not present in the set. One possible explanation could be that the two separate methylase gene fragments (2238 and 2241) are in fact active if both are expressed or alternatively a third gene is necessary to activate 2241. In any event this is a cautionary note that should be mentioned. It might even be worth repeating the experiment with both ORFs included in the plasmid. If that turns out to be the explanation, then this is a point that can be made very solidly within the paper.

The two genes 2238 and 2241 are separated by a transposon. In a related strain we found a methyltransferase in a single ORF (without the transposase) we tested it with the same motif box but could not detect any activity. When revising the data we observed indications that CTCCG was recognized by the methyltransferases MothHH_00692, and this has now been commented on in the results.

From Results: "The motif CTCCG was not properly identified by any of the cloned methyltransferases or the tested combination of TRD and methyltransferases, given the low detection rate on the genome-sequencing (table 1) this was expected. However, the motifs of MothHH_00692 indicated that it could have activity towards the CTCCG motif (extended motif including the motifs STSCN)."

13. Line 178-180. The motif_summary.csv file available to this reviewer showed only 3 motifs that were completely uninterpretable. If ORF 18590 really is the m5C methylase CCWGG then this result is not surprising since contrary to the earlier statement (line 140) 250 fold coverage does not guarantee correct identification of an m5C motif. This result could be used to emphasize this problem, which is not well appreciated generally, and apparently not by these authors.

The data presented on Awo_c18590 was used to emphasize the problem determining m5C using PacBio.

From Results: "Activity of Awo_c18590 resulted in recognition of four motifs, all with relative low modification QV (32-52), associating the motifs to the methylome, the activity could be assigned to motif CCWGG. Despite a coverage of 8514 (>250) modification QV was low and the motif extended, this indicates that detection of m5C by PacBio is incomplete."

14. Line 185. Two of these methylase genes (34920 and 35180) are encoded on prophages, which might account for their apparent inactivity. They may need additional genes to activate them or their recognition sequences may not be present since prophage methylases are not usually expressed in the genome and so no motifs would be present in the original genome analysis.

This has now been elaborated in the manuscript.

From results: "It was observed that two of the genes Awo_c34920, Awo_c35180 are encoded on prophages, and the genes might therefore not be active in the genome as prophage-encoded genes generally are transcriptionally inactive until the prophage is excised⁴⁶."

15. Lines 233-249. Having attempted to clone many genes from thermophilic organisms into a mesophile such as E. coli, I would be a little more cautious in claiming that they are usually expressed with little difficulty. In some cases additional genes are necessary for activity.

The statement has been moderated.

From Discussion: "Others have reported successful heterologous expression of thermostable methyltransferases in mesophilic systems^{52, 53}, however it must be stressed that this observed compatibility of thermostable methyltransferases may not be general. Activity detection of the thermophilic Type I..."

16. Line 254. This requirement for 250-fold coverage for m5C detection should not be repeated here. PacBio is just not suited to m5C detection, although occasionally it can give some indication of an active m5C methylase. There are other better methods for this.

The repetition has been deleted, and through out the manuscript we are more critical towards utilizing PacBio for m5C detection.

17. Line 262. While it is true that Clark et al. only included Type II and III methylases, a later paper (ref 23 in this manuscript) did include a number of Type I systems.

This has been corrected, acknowledgement has been given. However, it should be noted that this section has been significantly rewritten.

18. Line 272-274. The Clark et al. paper examined the complete bacterial genomes in which the methylase genes were cloned, not just the plasmids. This provided a very large sequence space in which the motifs could occur.

Thanks for the corrections, it has been corrected.

From discussion: "Heterologous expression of methyltransferases (from plasmid) and subsequently assessing the methylation patterns by SMRT sequencing of both plasmids (various plasmids are needed as methylation sites needs to be present) and genomes of the expressing hosts^{2, 39, 58} has been the method of choice"

19. Lines 287-289. While this is true and I have no problem with it being included, I would note that we are going to stop making predictions as flimsy as some of the current ones now that we are establishing a gold standard set of methylases with experimentally verified recognition sequences. The results from this paper help in that endeavor and I hope others will decide to use it and generate yet more experimental data.

Some of the points raised above could usefully be included in the Discussion (or expanded in the Results), since it is always helpful to the user to know why apparent failures might show up.

Rich Roberts

REVIEWERS' COMMENTS:

Reviewer #2 (Remarks to the Author):

The authors have responded well to my concerns.